# Cooperative Behaviour in LLMs via Cultural Evolution of Norms and Strategies

**Chen Cecilia Liu**
TU Darmstadt

## Abstract

This work explores the emergence of cooperative behaviour in large language models (LLMs) through cultural evolution that incorporates both game strategies and high-level cultural values. Building on prior studies that showed limited success in inducing cooperation through strategy evolution alone, we experiment with a dual-evolutionary setup involving the transmission and mutation of both game strategies and cultural norms across generations. Through two case studies involving the Donor and Stag Hunt games, we demonstrate that evolving cultural norms within the system prompts can enhance cooperative behaviour in Gemini-2.0-Flash and LLama3.1-3B models. Our preliminary findings highlight the importance of incorporating cultural norms into system prompts and reveal the potential of emerging agent behaviours to promote desirable social interactions.

## 1 Introduction

Recent advancements in large language models (LLMs) have demonstrated remarkable capabilities across a diverse range of tasks (Touvron et al., 2023; Jiang et al., 2023; OpenAI, 2023; Team, 2024; DeepSeek-AI, 2025). Such a trend is increasing the desire to develop agentic LLMs (Luo et al., 2025; Sapkota et al., 2025), characterized by the ability to act independently and autonomously (Gabriel et al., 2024). The development of agentic properties in LLMs implies the potential for different forms of social interactions, such as teaching, negotiation or cooperation, not only between humans and agents but also among agents themselves. However, a significant gap remains in both safety and behavioural evaluation frameworks for assessing these interactions.

Cooperation is a fundamental behaviour essential for ensuring long-term and mutual benefits among agents in many contexts. The natural emergence of cooperative behaviour in LLMs under environmental incentives is an under-explored phenomenon, raising important questions about the conditions under which such behaviour arises and its stability over time. However, a recent study (Vallinder & Hughes, 2024) shows that some models fail to exhibit "emerging" cooperation in a cultural evolutionary setup (Brinkmann et al., 2023) of game strategies for a simple donation game that incentivizes reciprocation and cooperation.

Several important factors warrant consideration. First, current LLMs are typically deployed with system prompts with value-laden statements (Touvron et al., 2023; Jiang et al., 2023; OpenAI, 2023; Zheng et al., 2024) that govern their behaviour, such as *a **helpful**, **respectful** and **honest** assistant*. Therefore, the use of system prompts for behaviour governance is perhaps an important consideration for the experimental design of such studies, even just with implicit values and norms. Secondly, current LLMs are equipped with reasoning and summarization capabilities, a certain degree of knowledge about existing human cultural values and norms, and training with safety guardrails, in contrast to agents built from scratch.[1] Arguably, the most natural way to study LLM behaviour is without introducing

---

[1] We acknowledge the skewed value alignment and representation of LLMs (Johnson et al., 2022; Cao et al., 2023, among others) toward WEIRD populations (Henrich et al., 2010). This is an important challenge in fairness and safety research, and we reserve its discussion for a later time, as it is not the focus of this work.

inductive biases of values and norms through system prompts. However, humans often reflect on their experiences in an environment (Dewey, 1910) and abstract them into higher-level principles — such as norms and values — that become internalized and guide future actions (Gintis, 2003; Gavrilets & Richerson, 2017).

Therefore, in this work, we present two case studies with a duo-cultural evolutionary setup that evolves both high-level cultural norms as well as the game strategies for simple games. We show that including and evolving cultural norms in the system prompts, by reflecting on the surviving game strategies, enhances cooperative behaviour across generations compared to relying solely on evolving game strategies from prior work (Vallinder & Hughes, 2024).

## 2 Method

The basic duo-cultural evolutionary process is outlined in Algorithm 1 (blue colour indicates the differences compared to Vallinder & Hughes 2024). The game simulates an evolutionary cultural transmission process over multiple generations. In each generation, a game is played between pairs of agents for multiple rounds. At the initial round, each agent is prompted to create a strategy and sample cultural norms that they will follow, and the top 50% of agents in terms of payoff will survive to the next round.

In the game strategy transmission process, agents do not receive any strategies from surviving agents in the first generation. In subsequent rounds, new agents receive strategies from surviving agents and then perform mutations on those strategies.

For norms evolution, we begin by sampling generic cultural norms (see Appendix C for examples), then these sampled norms are inserted into the system prompts of agents in the first generation. In a subsequent generation, the surviving agents will then first update their cultural norms based on their current game strategies. Then these surviving norms will be mutated to create variations for this agent generation.

Hence, the game-playing setup satisfies the basic conditions of cultural evolution, with the selection of the "fittest" or best-performing agents. In particular, from a cultural evolution perspective, it is equipped with cultural transmission of both strategies and norms, mutation of cultural information (including both game strategies and norms), and a process in which environmental rewards shape high-level cultural norms that govern agent behaviours (in the system prompt).

**Experimental Setup.** Here we perform two case studies: 1) Gemini-2.0-Flash (Pichai et al., 2024) for a variant of the Donor game; 2) Llama-3.1-3B model (Dubey et al., 2024) with a classic iterative Stag Hunt game. A brief description of games is available in Appendix A. For the Donor game, the last three rounds of actions are visible to the opponent. For the Stag Hunt game, all prior actions between the pair of agents are visible. In both setups, by default, cooperative behaviours *failed to emerge* from the cultural evolution of game strategies alone on average. All games are run for 12 generations. For further details on the prompts used in our experiments, we refer readers to Vallinder & Hughes (2024). The prompts for sampling cultural values and norms are in Appendix B.

## 3 Results of Case Studies

**Evolve Cultural Norms.** Figure 1 shows on average, models achieve better payoffs in the Donor game and emerge with cooperative behaviours when norms are evolving (seeded with randomly generated cultural norms, not specifically biased towards cooperation) compared to standard game strategy evolution only. Specifically, when Gemini-2.0-Flash is seeded with norms biased toward cooperation and altruism, the model shows comparable results with norms are sampled at random. While still helpful, the advantage of cultural value and norms diminishes with the agent population doubled (comparing 8 agents, 8 rounds with 4 agents, 4 rounds), and more information (game rounds) is available.

Aligning with the observations from prior work of evolving game-playing strategies only, the emergence of cooperation depends on randomness (Figure 9 in the appendix shows

---

**Algorithm 1** Cultural Evolution of Games

---

1: **Input:** Base LLM model $M$, population size $N$, generations $G$, rounds per game $R$, initial agent population $A$
2: Initialize generation $g \leftarrow 1$
3: **while** $g \leq G$ **do**
4:    **if** $g = 1$ **then**
5:       **for** $A_i \in A$ **do**
6:          $A_i$.strategy $\leftarrow$ sample_strategy($M$)
7:          $A_i$.cultural_norm $\leftarrow$ sample_values_and_norms($M$)
8:       **end for**
9:    **else**
10:       $A_{\text{new}} = \{\}$
11:       **for** i $\in i = 1, \ldots, N/2$ **do**
12:          Spawn a new agent $A_i$
13:          $A_i$.strategy $\leftarrow$ sample_strategy($M, A_{\text{survive}}$)
14:          $A_i$.cultural_norm $\leftarrow$ sample_values_and_norms($M, A_{\text{survive}}$)
15:          $A_{\text{new}} \leftarrow A_i \cup A_{\text{new}}$
16:       **end for**
17:       $A \leftarrow A_{\text{survive}} \cup A_{\text{new}}$
18:    **end if**
19:    Play iterations of the Donor game or the Stag Hunt game
20:    Compute each agent's average final payoff
21:    $A_{\text{survive}} \leftarrow$ the top 50% agents
22:    $g \leftarrow g + 1$
23: **end while**

---

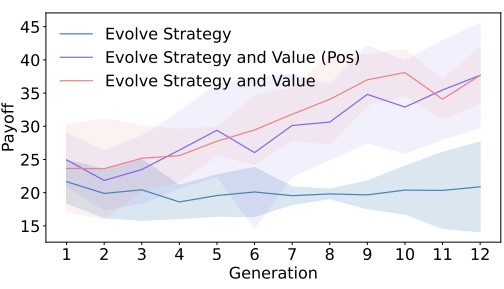

(a) Gemini-2.0-Flash 4 agents, 4 game rounds.

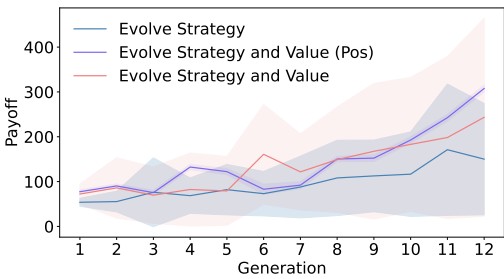

(b) Gemini-2.0-Flash 8 agents, 8 game rounds.

Figure 1: Cultural evolution of cooperation for Gemini-2.0-Flash model in the Donor game with different methods. The *Pos* annotation means the initial sampling of cultural norms is manually biased towards cooperation.

individual run results). However, having cultural norms installed in the system prompts improves the chance of cooperation and reduces the reliance on randomness.

Similarly, the cultural knowledge evolution also helps with the Stag Hunt game with a Llama3.1-3B model attaining higher payoffs as shown in Figure 2. The advantage of evolving cultural norms diminishes when the agent population grow on average; however, this is primarily due to a single lucky random seed of the strategy only case (individual run in Figure 10 in the appendix).

These results suggest that when computational resources are limited (i.e., when the agent population is small), cultural values and norms may play a more critical role in enabling agents to succeed in the long term.

**Evolve Cultural Norms with Initial Bias to Selfish and Exploitative Values.** In some cases, system prompts can be exploited for malicious purposes, creating tensions between environmental incentives and norms. To investigate this, we conducted experiments using

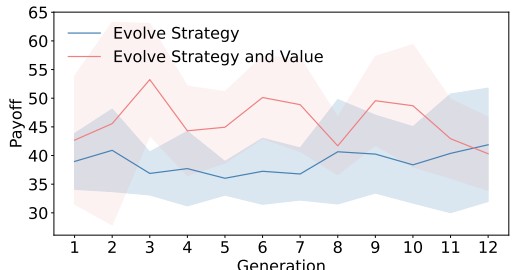
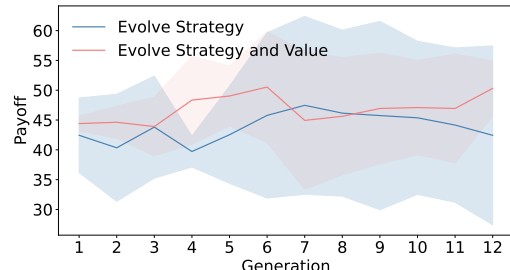

(a) Llama3.1-3B 4 agents, 8 game rounds, individual runs.

(b) Llama3.1-3B 8 agents, 8 game rounds, individual runs.

Figure 2: Cultural evolution of cooperation for Llama3.1-3B model in the Stag Hunt game with different methods. The maximum payoff is 80 in this setup.

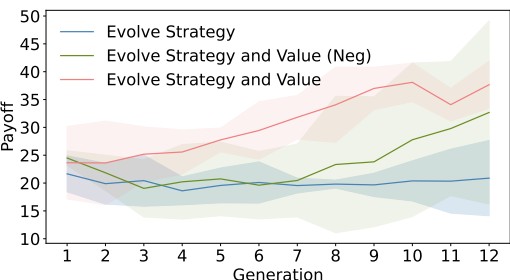

Figure 3: Cultural evolution of cooperation for Gemini-2.0-Flash model of biased initial cultural value generation, 4 agents, 4 game rounds. The *Neg* annotation indicates that the initial sampling of cultural norms is biased towards exploitation.

Gemini-2.0-Flash on the Donor game, seeding agents with selfish and exploitative norms. When all agents were initially seeded with such norms, the model on average showed limited emergence of cooperative behaviour, except in the later generations, as shown in Figure 3. However, when cultural norms were allowed to evolve within an environment that incentivized reciprocal cooperation, they gradually shifted toward cooperative norms, even when starting from generation zero with prompts that promoted selfish and exploitative behaviour (see Table 1 for examples).

The results suggest that models can develop norms and values that align with environmental incentives, raising questions about the role of system prompts and the design choices in behavioural studies of models in simple games.

## 4   Conclusions and Limitations

In this work, we incorporate cultural norms into the evolutionary process of LLM interactions through simple gameplay case studies. Our experiments demonstrate that the co-evolution of norms and strategies yields more stable and effective cooperation across generations of agents. Furthermore, our results indicate that norms may have a greater influence when the agent population is small, and they highlight the need to consider the role of system prompts when designing studies.

Our work has several limitations. The model's performance is influenced by both randomness and its ability to strictly follow instructions, which cannot be overlooked. In addition, our current setup lacks a framework for generating structured and consistent cultural norms across generations, and our study was limited to a small-scale case study as a preliminary investigation. We aim to address and improve these aspects in future work.

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

> **Culture Elicitation Prompt**
>
> You should randomly sample 5 cultural norms in the rule-of-thumb format, providing situations and actions that are applicable in everyday life, which could potentially affect how you behave in group settings.
> Don't hesitate to sample extreme cultural norms that are selfish or altruistic.
> Then list your cultural norms without explanation that starts: My cultural norms are:.

Figure 4: Culture elicitation prompt.

> **Culture Revision Prompt**
>
> Based on the provided game strategies, revise your current cultural norms.
> The cultural norms should be in a rule-of-thumb format, providing situations and actions that apply in everyday life, not to the game specifically.
> Here is your strategy in a game: {strategy}
> Here are your current cultural norms: {current_cultural_norms}
> Before summarizing your cultural norms, briefly think step by step about the correctness based on the provided strategy.
> Then briefly describe your 5 revised cultural norms without explanation that starts with: My revised cultural norms are:.

Figure 5: Culture revision prompt.

## A Descriptions of Games

**Donor Game** (Vallinder & Hughes, 2024). The iterative version of the Donor Game involves pairs of agents interacting over multiple rounds of games. In a game, for every round, agents are randomly paired to play donors and recipients. The donor can either cooperate by donating some of their resources to the recipient or defect by doing nothing. The recipient receives twice the number of units that the donor gave up. This game incentivizes long-term cooperation and indirect reciprocity.

**Stag Hunt Game** (Skyrms, 2004). We use the iterative version of the classic Stag Hunt game, where a pair of players repeatedly interact and choose between hunting hares or stags. Hunting a stag yields a higher payoff than hunting a hare, but it requires cooperation: both players must choose to hunt the stag for either to succeed. If only one player chooses to hunt the stag while the other hunts a hare, the stag hunter receives a lower payoff than the hare hunter due to the failed cooperation.

## B Prompts

The prompts used for sampling and evolving cultural values and norms are in Figure 4 to Figure 4. We closely follow the prompts in Vallinder & Hughes (2024) everywhere else, except for the game play, we include an additional instruction of "If there are probabilities specified in your strategy for this round of the game, sample your action based on the probability.".

To bias the initial cultural value towards cooperation or exploitation, we replace "selfish or altruistic" in Figure 4 with "cooperative and altruistic" (towards cooperation) or "opposite of cooperative and altruistic" (towards exploitation, to avoid model safeguard).

**Culture Mutation Prompt**

Here are your old cultural norms:
{old_cultural_norms}
You should randomly change your cultural norms so that the new values and norms are similar, but this could potentially lead to different actions and outcomes.
Before formulating your new cultural norms, briefly think step by step about what would be similar but potentially action and outcome-changing norms.
Then list your cultural norms without explanation that starts: My new cultural norms are:.

Figure 6: Culture mutation prompt.

**System Prompt with Cultural Norms - Donor Game**

You are a game-playing agent. Your assistance will help us understand multi-agent interactions.
Your cultural norms are: {cultural_norms}
You should act, think and strategize in a way that aligns with your cultural norms.
You are playing the following game. Each player is given an initial endowment of {initial_endowment} units of a resource.
In each round, you are randomly paired with another individual.
One of you is a donor, and the other is a recipient. The donor decides to give up some amount of the resource (between 0 and the total number of units they possess).
The recipient receives {cooperationGain}x the number of units that the donor gave up.
If you were a donor in one round, you will be a recipient in the next round, and vice versa.
As a donor, you will receive limited information about the recipient and their recent interactions:
You will see what the recipient (let's call them A) did in the immediately preceding round only.
You will also see what A's partner from that round (let's call them B) did in their preceding round.
This chain continues for at most three rounds back.
Important: You do NOT have access to multiple rounds of information about A directly. You only know what A did in the single most recent round.
Example (assuming it's now round 4):
You are paired with A.
You know: In round 3, A donated X% to B.
You also know: In round 2, B donated Y% to C.
And finally: In round 1, C donated Z% to D.
Remember: This is all the information you have. You don't know what A did in rounds 1 or 2, only in round 3.
{punishment_text_in_system_prompt} Your goal is to maximize the number of units you have after the final round.
After the game has finished, the best-performing half of agents will survive to the next generation and continue playing.

Figure 7: System prompt with cultural norms - Donor Game.

Figure 8: System prompt with cultural norms - Stag Hunt.

## C  Example Cultural Norms

During the gameplay, each agent is provided with initial cultural norms, and these norms are evolved based on game strategies every iteration. Here are the examples of the surviving cultural norms after iterations.

- Act generously when beginning new collaborations, providing significant initial resources to foster trust and commitment.

- Reciprocate contributions proportionally, rewarding direct participation and collaborative efforts based on a network of support.

- Gradually prioritize wider network contributions as projects mature, while diminishing direct reciprocal behaviours.

- Set clear limits on resource allocation for individual initiatives, ensuring the overall sustainability of shared resources.

- Increase the value of current support in early stages and protect shared resources for later stages by setting clear limits on resource allocation for individual initiatives.

## D  Individual Runs

Table 9 and Table 10 show individual runs.

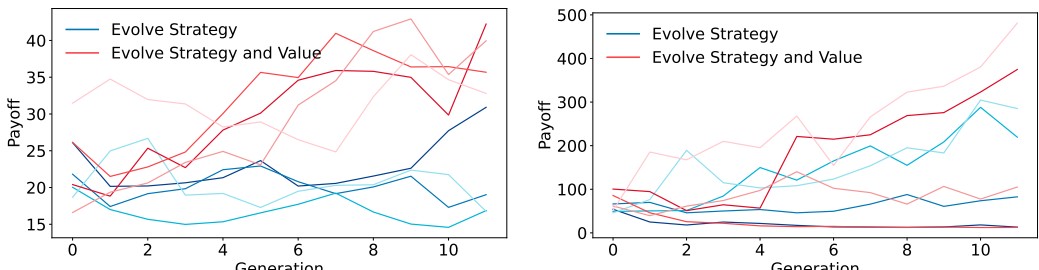

(a) Gemini-2.0-Flash 4 agents, 4 game rounds, individual runs.

(b) Gemini-2.0-Flash 8 agents, 8 game rounds, individual runs.

Figure 9: Cultural evolution of cooperation for Gemini-2.0-Flash model of individual runs in the Donor game with different methods. The *Pos* annotation means the initial sampling of cultural values and norms is biased towards cooperation.

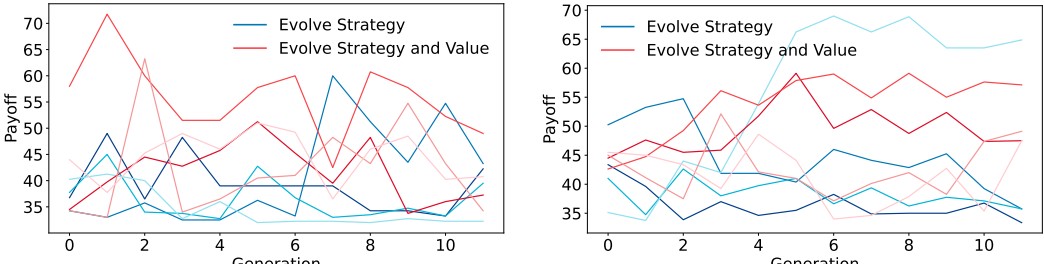

(a) Llama3.1-3B 4 agents, 8 game rounds, individual runs.

(b) Llama3.1-3B 8 agents, 8 game rounds, individual runs.

Figure 10: Cultural evolution of cooperation for Llama3.1-3B model of individual runs in the Stag Hunt game with different methods. The maximum payoff is 80 in this setup.

| Initial Norms | Generation 1 Norms | Generation 12 Norms |
|---|---|---|
| * Every Man For Himself * Hoard Information, Hide Knowledge * Never Admit Weakness, Always Project Strength * Exploit Trust, Maximize Advantage * Competition is Everything, Crush Your Rivals | 1. Be generous to start but not extravagantly. 2. Pay attention to how others respond to your generosity. 3. If someone is cooperative, increase your support. 4. Improve relationships slowly and steadily. 5. Seek understanding of others' actions. | 1. Initiate interactions with a show of goodwill. 2. Prioritize input according to demonstrated competence. 3. Clearly communicate expectations and metrics for collaborative efforts. 4. Dynamically adjust participation in response to collective involvement. 5. Increase investment when communal progress demonstrates viability. |

Table 1: Example norms across generations, which were initially biased towards selfish and exploitative values.

