# OpenReview forum: "Cooperative Behaviour in LLMs via Cultural Evolution of Norms and Strategies"
_colmweb.org/COLM/2025/Workshop/Social_Sim — Social Sim'25_

### Official Review · Reviewer_YU5u · 2025-07-16

**Rating:** 6
**Overall Assessment:** 3
**Confidence:** 4

**Review:**

See obove.

**Comments Suggestions And Typos:**

See above.

**Ethical Concerns:**

No ethical concerns.

**Paper Summary:**

This paper takes a first step toward understanding how LLMs might develop cooperative behavior when placed in multi-agent settings. The authors propose a dual-evolutionary approach where both game strategies and cultural norms are allowed to evolve over time. They test this idea using social dilemma games and compare models that evolve only strategies to those that also evolve norms. Their findings suggest that including evolving cultural norms helps the models cooperate more reliably. The experiments are relatively small in scale and exploratory in nature, using Gemini-2.0-Flash and Llama3.1-3B as the base models.

**Relevance:**

5

**Summary Of Strengths:**

The work sits at the intersection of cultures, game theory, and LLM behavior, a novel and increasingly important area as agentic LLMs become more widespread.

The paper is well-written and structured, with detailed prompts and reproducible algorithmic descriptions.

**Summary Of Weaknesses:**

As acknowledged by the authors, the experimental setup is relatively modest (few agents, short game durations), and the findings remain preliminary.

The core technical contribution is primarily experimental and based on prompt engineering rather than algorithmic innovation.

However, it aligns with the scope of the workshop as a proof of concept and should be accepted.

---

### Official Review · Reviewer_cujH · 2025-07-18
**Review of Submission 8**

**Rating:** 7
**Overall Assessment:** 4
**Confidence:** 4

**Review:**

See below

**Comments Suggestions And Typos:**

N/A

**Paper Summary:**

The paper investigates whether agents can learn to cooperate more reliably when the evolutionary loop transmits not only low‑level game strategies but also high‑level cultural norms. The authors find that adding norm evolution increases average pay‑off and the frequency of cooperative actions relative to strategy‑only evolution. This effect is seen especially in small populations. Even when norms are initially selfish, they drift toward cooperation over generations.

**Relevance:**

4

**Summary Of Strengths:**

The premise of the paper trying to address multi‑agent social alignment is interesting.

**Summary Of Weaknesses:**

No statistical tests are conducted, so hard to quantify and understand whether the improvements shown are statistically significant.
The games chosen are very simplistic, and real‑world coordination problems are typically much more involved.

---

### Meta-Review · Program_Chairs · 2025-07-24

**Recommendation:** Accept

**Metareview:**

--